# Impact of Anti-TNFα Treatment on the Humoral Response to the BNT162b2 mRNA COVID-19 Vaccine in Pediatric Inflammatory Bowel Disease Patients

**DOI:** 10.3390/vaccines10101618

**Published:** 2022-09-27

**Authors:** Kosuke Kashiwagi, Keisuke Jimbo, Mitsuyoshi Suzuki, Nobuyasu Arai, Takahiro Kudo, Toshiaki Shimizu

**Affiliations:** 1Department of Pediatrics and Adolescent Medicine, Juntendo University Graduate School of Medicine, Tokyo 113-8421, Japan; 2Department of Pediatrics, Juntendo University Faculty of Medicine, Tokyo 113-8421, Japan

**Keywords:** pediatric inflammatory bowel disease, BNT162b2 mRNA vaccine, humoral response, anti-TNFα, third vaccination

## Abstract

The efficacy of the COVID-19 mRNA vaccine, including the third vaccination in pediatric inflammatory bowel disease (PIBD) patients is not fully understood. This study aimed to evaluate the humoral immunogenicity of the BNT162b2 vaccine and the changes in durability until 20–28 weeks after the initial vaccine series in PIBD patients on immunosuppressive drugs. The safety of the initial vaccine series and the booster effect of the third vaccination were also evaluated. A single-center, prospective cohort study was conducted, and 63 participants (anti-TNFα: 11; non-anti-TNFα: 31; 5-ASA: 21), with a mean age of 15.2 (range 9.6–17.9) years, were enrolled. All PIBD patients were seroconverted, with no serious short-term AEs. PIBD patients on anti-TNFα had significantly lower antibody titers than those on other medications at all measurement points. Furthermore, antibody titers waned over time with anti-TNFα and were significantly lower at 20–28 weeks than at 3–9 weeks after a two-vaccine series. In all 10 patients (anti-TNFα: 5; non-anti-TNFα including 5-ASA: 5), the third vaccination led to antibody concentrations significantly higher than those at the same time point after the second vaccination. PIBD patients on anti-TNFα need to remain vigilant about COVID-19 even after two vaccinations, and a third vaccination may be considered.

## 1. Introduction

Coronavirus disease 2019 (COVID-19), caused by the novel coronavirus referred to as severe acute respiratory syndrome coronavirus 2 (SARS-CoV-2), emerged in late 2019 and subsequently spread worldwide [1,2]. Vaccination against SARS-CoV-2 is important for protection against SARS-CoV-2, and the first COVID-19 messenger RNA (mRNA) vaccine (Pfizer BioNTech: BNT162b2) was approved for those over 18 years of age in December 2020 [2]. Regarding children, the number of children with COVID-19 has risen nationwide. Most children with COVID-19 have a mild disease [3,4], but a risk of severe disease has been reported in children younger than 2 years and those with underlying disease [5,6]. Thus, the mRNA vaccination has been approved for children [7], and it has been licensed for emergency use authorization (EUA) in children aged 5–11 years in the United States since October 2021 [8]. In addition, at the time of writing, the recommendation of the Centers for Disease Control and Prevention (CDC) is that children should receive a booster shot 5 months after the second vaccination.

Patients with inflammatory bowel disease (IBD) often require immunosuppressive therapy. However, clinical trials that have shown the SARS-CoV-2 mRNA vaccine’s efficacy and safety in the general population excluded patients on immunosuppressive drugs [9]. Thus, the immunogenicity of the BNT162b2 mRNA COVID-19 vaccine in IBD patients on immunosuppressive drugs has received attention. Several recent studies of adult IBD patients have shown a reduction of the humoral response to the BNT162b2 mRNA vaccine in persons treated with anti-tumor necrosis factor (TNF) α compared with those not receiving anti-TNFα or healthy controls [10,11,12,13].

However, in pediatric IBD (PIBD) patients, only a few studies have shown a reduction of the serological response against the BNT162b2 mRNA vaccine in those treated with biologic therapy, especially anti-TNFα [14,15,16]. In one study, antibody titers were significantly lower in PIBD patients on infliximab combination therapy than in healthy adults 28 days after the first vaccination [14]. However, antibody titers in PIBD patients were evaluated over a short follow-up period of 3 months after the second vaccination in that study. In addition, another study compared serum antibody titers and the safety of the BNT162b2 vaccine in PIBD patients treated with anti-TNFα and non-anti-TNFα drugs [16]. This study also showed a reduction of the serological response to the BNT162b2 vaccine in PIBD patients on anti-TNFα, but this study was also a short-term observational study, with follow-up to 7 days after the second vaccination. Thus, the differences in long-term immunogenicity of the BNT162b2 vaccine of PIBD patients between those on anti-TNFα and those on other IBD drugs after the initial vaccine series remain unknown. Moreover, the booster effect in PIBD patients after the third vaccination is also unknown.

Therefore, this study aimed to determine whether humoral immunogenicity is altered and the changes in durability over time after the initial vaccine series in PIBD patients, especially those on anti-TNFα. In addition, the short-term adverse events (AEs) of the BNT162b2 vaccine and the booster effect of the third vaccination were also evaluated.

## 2. Materials and Methods

### 2.1. Study Design and Participants

A single-center prospective study was conducted between September 2021 and August 2022 to assess the serological responses of PIBD patients to the mRNA-based BNT162b2 COVID-19 vaccine. Eligible patients were PIBD patients under 18 years of age who had received two doses of the BNT162b2 vaccine (Pfizer/BioNTech). Patients who had confirmed COVID-19 or positive serology before the second vaccination were excluded. The PIBD patients were divided into three groups: those treated with anti-TNFα; those treated with other immunosuppressive agents (called non-anti-TNFα: thiopurine (azathioprine (AZA) or mercaptopurine (6-MP)) monotherapy, ustekinumab (UST) monotherapy and thiopurine combination therapy); and 5-aminosalicylic acid (5-ASA) alone.

### 2.2. Study Procedures

Serum separated from blood samples of participants, stored at −80 °C until analysis, was used, and the levels of binding immunoglobulin G (IgG) antibodies to SARS-CoV-2 spike (S) antigen were measured at 3–9 weeks, 10–16 weeks, and 20–28 weeks after the second vaccination. Antibody titers were compared among the three groups at each time point, and the changes in antibody titers over time were evaluated. In addition, antibody concentrations were measured 3–9 weeks after the third vaccination in patients who received the third vaccine dose and whose serum samples could be collected. In the comparison of antibody titers after the third vaccination, patients treated with anti-TNFα and those treated with non-anti-TNFα including 5-ASA were compared. Patients infected with SARS-CoV-2 after the second vaccination were investigated for the time of infection and the variant of SARS-CoV-2. Serum samples obtained after infection were excluded from the evaluation. The Roche Elecsys Anti-SARS-CoV-2 spike (S) electrochemiluminescence immunoassay (Roche Diagnostics International Ltd., Rotkreuz, Switzerland) was used to identify vaccination-specific antibody responses [17]. Concentrations of anti-SARS-CoV-2 S protein antibodies greater than 15 units (U)/mL on the Elecsys assay were associated with neutralization of at least 20%, with a positive predictive value of 99.10% (95% confidence interval 97.74–99.64) [18]; therefore, 15 U/mL was defined as the threshold of seroconversion.

To investigate short-term AEs, questionnaires regarding fever, local injection site reaction (pain, erythema, or swelling), and systemic AEs (fatigue, headache, muscle or joint pain, nausea, allergic reaction) within 7 days after the first and second vaccinations were included. Fever was assessed according to the following scale: 0: none (<37.5 °C) 1: 37.5–38 °C, 2: 38–38.4 °C, 3: 38.5–38.9 °C, and 4: ≥39 °C. In addition, the severities of local injection site reactions and systemic AEs were assessed according to the following scale: 0: none; 1: does not interfere with activity; 2: interferes with activity; and 3: prevents daily activity and/or requires hospitalization.

### 2.3. Statistical Analysis

All statistical analyses and visualizations were performed using GraphPad Prism ver. 9.1.2 (GraphPad Software, San Diego, CA, USA). For multiple comparisons, the Brown–Forsythe and Welch analysis of variance was followed by the Dunnett T3 correction. For comparisons of nonparametric variables, the unpaired non-parametric Mann–Whitney test was used. Paired continuous samples in the same groups were compared using the paired *t*-test. *p*-values < 0.05 were considered significant.

## 3. Results

### 3.1. Patients’ Characteristics

A total of 63 patients with a mean age of 15.2 (range 9.6–17.9) years were enrolled for this study, and the main characteristics of all participants are summarized in Table 1. All 63 patients were Japanese, 29 patients (46.0%) were female, and 43 patients (68.3%) had ulcerative colitis (UC). In the comparison of time from the beginning of the IBD treatment to the second vaccination, patients treated with 5-ASA had a significantly shorter time than patients treated with non-anti-TNFα (non-anti-TNFα (34 months) vs. 5-ASA (10 months), *p* < 0.05). No patients had IBD activity, and all patients were in clinical remission at the time of analysis. There were 145 serum samples for analysis of the serological response to the second vaccination (anti-TNFα: 27 samples in 11 patients; non-anti-TNFα: 72 samples in 31 patients; 5-ASA: 46 samples in 21 patients). No patients were treated with Prednisolone. None of the participants had any evidence of prior SARS-CoV-2 infection before the second vaccination.

### 3.2. Anti-SARS-CoV-2 S Antibody Concentrations after the Initial Vaccine Series

The geometric mean of anti-SARS-CoV-2 S protein antibody concentrations following the second vaccination was significantly lower in patients treated with anti-TNFα than in those treated with non-anti-TNFα and 5-ASA at all measurement points (3–9 weeks: anti-TNFα (439 U/mL) vs. non-anti-TNFα (1702 U/mL) vs. 5-ASA (2096 U/mL), both *p* < 0.01; 10–16 weeks: anti-TNFα (188 U/mL) vs. non-anti-TNFα (1737 U/mL) vs. 5-ASA (2658 U/mL), both *p* < 0.01, respectively; 20–28 weeks: anti-TNFα (140 U/mL) vs. non-anti-TNFα (1650 U/mL) vs. 5-ASA (2233 U/mL), both *p* < 0.01, respectively). There were no significant differences in antibody titers between patients treated with non-anti-TNFα and 5-ASA (Figure 1).

### 3.3. Anti-SARS-CoV-2 S Antibody Concentrations at Different Time Points after the Initial Vaccine Series

All PIBD patients were seropositive until 20–28 weeks after the second vaccination, except for one patient on anti-TNFα who became seronegative at 20–28 weeks (9.0 U/mL). In patients treated with anti-TNFα, there was no significant difference in antibody concentrations at 3–9 and 10–16 weeks, but antibody titers waned over time, and those at 20–28 weeks were significantly lower than those at 3–9 weeks (3–9 weeks (439 U/mL) vs. 10–16 weeks (188 U/mL) vs. 20–28 weeks (140 U/mL), *p* = 0.05 and *p* < 0.01, respectively) (Figure 2). On the other hand, in both patients treated with non-anti-TNFα (3–9 weeks (1702 U/mL) vs. 10–16 weeks (1737 U/mL) vs. 20–28 weeks (1650 U/mL), *p* = 0.99 and *p* = 0.99, respectively) and those treated with 5-ASA (3–9 weeks (2301 U/mL) vs. 10–16 weeks (2658 U/mL) vs. 20–28 weeks (2233 U/mL), *p* = 0.92 and *p* = 0.99, respectively), antibody concentrations were sustained over time.

### 3.4. Anti-SARS-CoV-2 S Antibody Levels after the Third Vaccination

Overall, 23 of 63 participants (36.5%) received a third vaccination, and 10 serum samples were collected and evaluated to analyze the serological response to the third vaccination (anti-TNFα: 5 samples; non-anti-TNFα: 4 samples; 5-ASA: 1 sample). In the comparison of antibody titers after the third vaccination, patients treated with anti-TNFα had significantly lower antibody levels (anti-TNFα (4896 U/mL) vs. non-anti-TNFα including 5-ASA (30,900 U/mL), *p* = 0.0079), the same as those after the second vaccination (anti-TNFα (361 U/mL) vs. non-anti-TNFα including 5-ASA (1223 U/mL), *p* < 0.01) (Figure 3). However, all patients had significantly higher antibody concentrations 3–9 weeks after the third vaccination than those at the same time point after the initial two vaccine series, regardless of the IBD treatment: anti-TNFα (3–9 weeks after the initial two vaccine series (361 U/mL) vs. 3–9 weeks after the third vaccine dose (4896 U/mL), *p* < 0.01); non-anti-TNFα including 5-ASA (3–9 weeks after the initial two vaccine series (1223 U/mL) vs. 3–9 weeks after the third vaccine dose (30,900 U/mL), *p* < 0.01) (Figure 3). Furthermore, one patient on anti-TNFα who became seronegative (9.0 U/mL) at 20–28 weeks after the second vaccination converted to seropositive, with antibody titers of 3120 U/mL. The average increase in antibody titers was 15.4 (5.7–23.2)-fold and 25.9 (18.2–39.5)-fold in patients treated with anti-TNFα and non-anti-TNFα including 5-ASA, respectively.

### 3.5. Short-term AEs after the First and Second Vaccinations

The frequency and extent of short-term AEs after the first and second vaccinations are summarized in Appendix A). Of the local injection site reactions, local pain was the most common and occurred in 81.0% and 79.4% of all PIBD patients after the first and second vaccinations, respectively. Of the systemic AEs after the first and second vaccinations, fatigue was the most common and occurred in 38.1% and 49.2% of all PIBD patients, respectively. The extent of the local injection site reaction and systemic AEs in patients on anti-TNFα were mild and comparable to the other two treatment groups. Patients on anti-TNFα tended to have higher, but not significantly higher, fevers than other patients after both the first and second vaccinations (Appendix A). In all patients, there were no serious gastrointestinal AEs, and none suffered from IBD exacerbation following the first and second vaccinations during the observational period.

### 3.6. Breakthrough Infection

A total of seven participants (11.1%) had histories of symptomatic COVID-19 after the initial two-vaccine series, and serum samples collected after COVID-19 were excluded from the analysis. These patients had mild symptoms, and none were hospitalized due to COVID-19. In addition, only one patient treated with non-anti-TNFα had a marked increase in antibody titers, to 22,900 U/mL, at 20–28 weeks after the second vaccination without COVID-19 symptoms. This patient was considered to be asymptomatic with COVID-19, and the serum samples were excluded from analysis after that (her serum sample after the third vaccination was also excluded from the analysis). All symptomatic patients were infected from January 2022 to May 2022, when the omicron variant (B.1.1.529) was prevalent in Japan. All participants vaccinated up to the third vaccination had no COVID-19 after the third vaccination within the observation period.

## 4. Discussion

Many studies have demonstrated the humoral response to the BNT162b2 vaccine and its efficacy in adult IBD patients on immunosuppressive therapy, but in PIBD patients, only a few reports have studied the humoral response to the BNT162b2 vaccine [14,15,16]. The present study showed the efficacy of the BNT162b2 vaccine in PIBD patients by showing that all patients who completed two vaccine doses were seroconverted, even those treated with anti-TNFα. On the other hand, patients on anti-TNFα had lower humoral responses to the BNT162b2 vaccine, and antibody titers of such patients decreased more rapidly over time than those on other IBD medications. However, the third vaccination was shown to boost antibody titers and convert seronegative patients to seropositive, even those on anti-TNFα who had become seronegative.

The present findings showed that all PIBD patients on IBD treatment, even those on anti-TNFα, became seropositive following the initial vaccine series. These results were comparable to the previous study of adult IBD patients, which showed that most adult IBD patients, including those treated with anti-TNFα, became seropositive after the initial two-vaccine series [19]. On the other hand, even in IBD patients who were seropositive after the second vaccination, breakthrough infections can occur. However, the systematic review suggested that the pooled relative risk of breakthrough infections in IBD patients was comparable to that in healthy control subjects [19]. In addition, it has been reported that patients on anti-TNFα who have completed their initial two-vaccine series rarely develop severe COVID-19, similar to the general population [20]. The present findings also showed that all cases with breakthrough infections after the second vaccination were mild cases that did not require hospitalization and were infected with omicron variants with a limited protective effect of two vaccine doses of the BNT162b2 vaccine [21]. These results suggest the effectiveness of protection against SARS-CoV-2 in PIBD patients who seroconvert after two vaccinations and encourage PIBD patients to be vaccinated.

Although the effectiveness of two vaccinations was confirmed in the present study, the current findings also showed that, in PIBD patients on anti-TNFα, antibody concentrations were lower and attenuated more rapidly over time than those treated with other IBD medications after the initial two-vaccine series. In adult IBD patients, those on anti-TNFα have been reported to have lower antibody titers than those on non-anti-TNFα and healthy controls [10,11,13]. PIBD patients have also shown similar trends in short-term follow-ups [14,16]. In addition, several studies have reported that, in adult IBD patients on anti-TNFα, antibody titers decrease more rapidly after the second vaccination than in those on other IBD medications [22,23,24]. These studies reported results similar to those in the present study.

However, concerning the decrease in antibody titers, it is unclear whether antibody titers alone reflect neutralization or protection against SARS-CoV-2 [25]. As a protective response to SARS-CoV-2, the T-cell response is increasingly recognized as an essential component in addition to the humoral response [19], and a previous study reported that adult IBD patients had T-cell responses similar to healthy controls after the initial two-vaccine series [25]. In addition, it has been reported that T-cell responses could be enhanced by anti-TNFα therapy [26]. These results suggested that the protective function against SARS-CoV-2 may be preserved in PIBD patients on anti-TNFα with attenuated antibody responses. However, not all humoral non-responders showed a cellular response [27], and adult IBD patients on anti-TNFα could fail to mount a T-cell response [24]. Thus, antibody titers remain crucial for protecting against COVID-19, and PIBD patients on anti-TNFα whose antibody concentrations are low and decayed should pay more attention to COVID-19 after the initial vaccine series. In addition, they require the third vaccination earlier than those treated with other IBD medications to reinforce the humoral response against SARS-CoV-2.

Concerning the third vaccination, it has been suggested that all IBD patients on immunosuppressive therapy aged 12 years or older should receive a third vaccination within 8 weeks after the second vaccination [28]. Some evidence has suggested that adult IBD patients who were seronegative after the initial vaccine series seroconverted after the third vaccination [29,30]. However, there were limited studies describing the response to the third vaccination, even in adult IBD patients. In the present study, although the sample size of PIBD patients who received the third additional vaccination was too small to make any conclusions, the third vaccination can lead to sufficient antibody titers and convert seronegative PIBD patients to seropositive, even those on anti-TNFα. These results suggest the efficacy of the booster effect of the third vaccination in PIBD patients regardless of their IBD treatment, and the present findings can be essential evidence to recommend the booster vaccination aggressively, especially for PIBD patients on anti-TNFα. In addition, the neutralization efficiency against the omicron variant was reported to be higher after the third vaccination than after the second vaccination in healthy adults [31]. Although it is difficult to draw any conclusions due to the small sample size and short-term follow-up, PIBD patients, including those on anti-TNFα, who completed the third vaccination had no COVID-19 during the observation period, suggesting the efficacy of the third vaccination for the omicron variant, especially B.1.1.529.

Some IBD patients are hesitant to receive the vaccine due to fear of AEs [32,33]. However, previous studies have reported that IBD was not associated with severe short-term AEs, even with biologics [10,34,35]. On the other hand, regarding fever in the present study, although the sample size was too small to make definitive conclusions, patients on anti-TNFα tended to have higher fevers than other patients. One previous report showed that anti-TNFα was associated with severe systemic AEs, including fever [36], therefore, the present findings suggest that PIBD patients on anti-TNFα are more likely to have higher fevers. However, at least there were no significant differences in the severity of AEs, and no severe short-term AEs occurred in PIBD patients in the present study. In addition, concerning the exacerbation of IBD, some studies reported that the frequency of IBD flares after vaccination was low [20,36]. Reassuringly, a previous study of adult IBD patients also reported no IBD exacerbation was observed within 6 months after the initial vaccination, even in IBD patients with activity at baseline [37]. The present findings showed that all PIBD patients had no IBD exacerbation during the observation period, and these results are consistent with previously published studies in adults. The effect on IBD activity for longer periods of time after vaccination is unknown, and studies for more long-term observation are required, but our results may support PIBD patients in getting vaccinated comfortably.

Finally, as mentioned before, antibodies will play an important role in defense against SARS-CoV-2 infection. However, the antibody-induced exacerbation of infection by some viruses, such as the dengue virus, has been reported as an antibody-dependent phenomenon (ADE) [38]. In SARS-CoV-2, a non-canonical, Fc-receptor-independent ADE has been reported, in which antibodies against a specific site on the N-Terminal Domain of the spike protein directly enhance the binding of ACE2 to the spike protein, increasing the infectivity of SARS-CoV-2 [39]. In this study, there were no cases of severe COVID-19 in PIBD patients after the initial vaccine series, but the risk of exacerbation of infection by ADE should always be considered when recommending vaccination.

There are several limitations to the present study. First, the small sample size in a single center makes it difficult to draw any definitive conclusions, especially about the efficacy of the third vaccination. Second, the differences in the assays used to measure the anti-SARS-CoV-2 S protein antibody and the different definitions of seroconversion thresholds may make direct comparisons with the previously reported antibody response data difficult. Third, the age of PIBD patients predominantly included in the present study was 12–18 years, and the humoral response in PIBD patients aged 5–11 years could not be assessed. Fourth, this study included only Japanese PIBD patients, and therefore, differences in humoral response by race could not be considered. Finally, cell-mediated immunity is an essential component of protection against COVID-19. However, the present study focused only on humoral immunity; therefore, the protective function in PIBD patients on each IBD treatment could not be accurately evaluated and compared. Further investigation of humoral and T-cell responses with a larger number of samples and a wider range of target ages will be required to assess the immunogenicity of the BNT162b2 vaccine in PIBD patients. In addition, the impact of ADE due to antibodies produced by the vaccination in PIBD patients should be widely surveyed.

## 5. Conclusions

The findings of the present study showed that two vaccine doses resulted in seroconversion in all PIBD patients, and these vaccinations were safe. However, the humoral response to the BNT162b2 vaccine in PIBD patients on anti-TNFα was attenuated, and antibody titers waned rapidly, with the possibility of becoming seronegative. On the other hand, reassuringly, the booster effect was observed in patients who received the third dose of the vaccine, regardless of IBD treatment. Thus, PIBD patients on anti-TNFα should be more careful about COVID-19 even after the initial vaccine series, and they may receive the third vaccine as soon as possible compared to patients receiving other IBD treatments.

## Figures and Tables

**Figure 1 vaccines-10-01618-f001:**
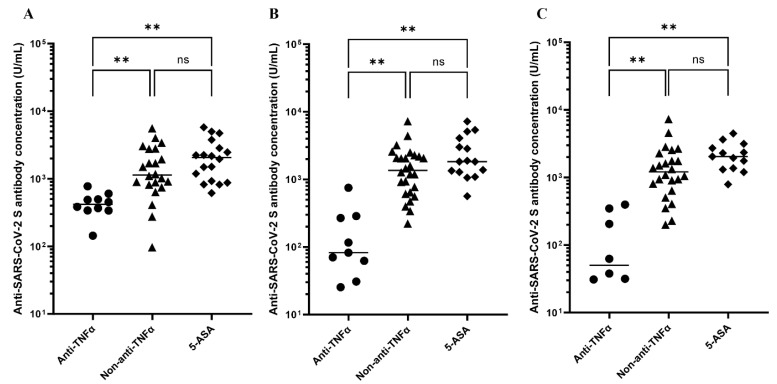
Anti-SARS-CoV-2 S antibody concentrations (U/mL) after the initial two-vaccine series. (**A**) Antibody titers of PIBD patients 3–9 weeks after the initial two-vaccine series. (**B**) Antibody titers of PIBD patients 10–16 weeks after the initial two-vaccine series. (**C**) Antibody titers of PIBD patients 20–28 weeks after the initial two-vaccine series. TNF = tumor necrosis factor, 5-ASA = 5-aminosalicylic acid. For multiple comparisons, Brown-Forsythe and Welch analysis of variance followed by the Dunnett T3 correction was used, and *p*-values < 0.05 were considered significant. ** indicates *p*-values < 0.01, and ns indicates not significant.

**Figure 2 vaccines-10-01618-f002:**
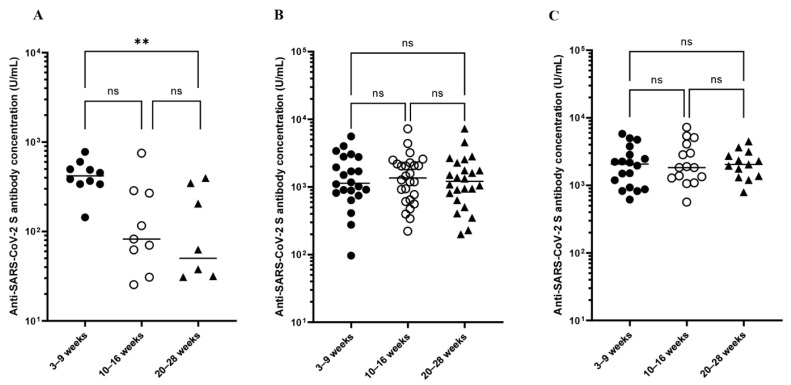
Anti-SARS-CoV-2 antibody concentrations (U/mL) at different time points after the initial two-vaccine series. (**A**) Antibody titers of patients treated with anti-TNFα decrease over time, and those at 20–28 weeks are significantly lower than those at 3–9 weeks. (**B**) Antibody titers of patients treated with non-anti-TNFα are sustained over time. (**C**) Antibody titers of PIBD patients treated with 5-ASA are sustained over time. TNF = tumor necrosis factor, 5-ASA = 5-aminosalicylic acid. For multiple comparisons, Brown-Forsythe and Welch analysis of variance followed by the Dunnett T3 correction was used, and -values < 0.05 were considered significant. ** indicates *p*-values < 0.01, and ns indicates not significant.

**Figure 3 vaccines-10-01618-f003:**
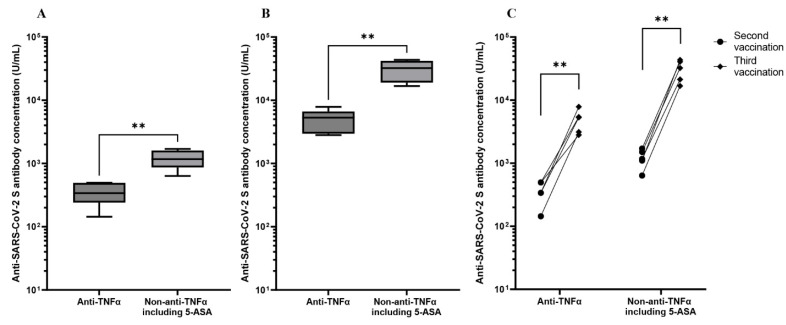
Serological response to the third vaccination. Ten patients were evaluated for the response to the third dose (anti-TNFα: 5, non-anti-TNFα including 5-ASA: 5). (**A**) Antibody titers in patients treated with anti-TNFα are significantly lower than those treated with non-anti-TNFα including 5-ASA 3–9 weeks after the second vaccination. (**B**) Antibody titers in patients treated with anti-TNFα are significantly lower than those treated with non-anti-TNFα including 5-ASA 3–9 weeks after the third vaccination. (**C**) Antibody titers 3–9 weeks after the third vaccination are significantly higher than those at the same point after the second vaccination, regardless of IBD medications. TNF = tumor necrosis factor, 5-ASA = 5-aminosalicylic acid. For comparisons of different vaccine groups in panels (**A**,**B**), the unpaired non-parametric Mann–Whitney test was used. For comparisons at different time points of the same vaccine groups in panel (**C**), paired two-tailed t-tests were used. For both tests, *p*-values < 0.05 were considered significant. ** indicates *p*-values < 0.01.

**Table 1 vaccines-10-01618-t001:** Participants’ baseline characteristics.

Variable		Overall (*n* = 63)	Patients on Anti-TNFα (*n* = 11)	Patients on Non-Anti-TNFα (*n* = 31)	Patients on 5-ASA (*n* = 21)
Age (years)		15.2 (9.6–17.9)	13.3 (9.6–17.9)	16.1 (12.3–17.9)	14.6 (11.8–17.9)
Sex	Female	46.0% (29/63)	45.5% (5/11)	53.3% (16/31)	36.4% (8/21)
	Male	54.0% (34/63)	54.5% (6/11)	48.4% (15/31)	61.9% (13/21)
Diagnosis	CD	31.7% (20/63)	63.6% (7/11)	33.3% (10/31)	13.6% (3/21)
	UC	68.3% (43/63)	36.4% (4/11)	67.7% (21/31)	85.7% (18/21)
Medications (Other than anti-TNFα and 5-ASA)	UST	17.5% (11/63)	0	36.7% (11/31)	0
	IM (AZA/6-MP)	49.2% (31/63)	54.5% (6/11)	83.9% (26/31)	0
Time from the beginning of the IBD treatment to the second vaccination (months)		21 (8–46)	18 (13–44)	34 (14–53)	10 (2–24)
Serum samples for analysis of antibody titers after the initial two vaccine doses	Total	145	18.6% (27/145)	49.7% (72/145)	31.7% (46/145)
	3–9 weeks	50	20.0% (10/50)	44.0% (22/50)	36.0% (18/50)
	10–16 weeks	50	18.0% (9/50)	52.% (26/50)	30.0% (15/50)
	20–28 weeks	45	17.8% (8/45)	53.3% (24/45)	28.9% (13/45)
Time from the initial two vaccine doses to the analysis of serum (days)	3–9 weeks	39 (21–63)	45 (21–63)	41 (21–63)	38 (21–63)
	10–16 weeks	93 (72–112)	94 (79–109)	93 (72–112)	94 (77–113)
	20–28 weeks	170 (140–196)	179 (168–194)	169 (141–196)	167 (140–194)
Patients with the third vaccination		36.5% (23/63)	54.5% (6/11)	41.9% (13/31)	19.0% (4/21)
Serum samples for analysis of antibody titers after the third vaccination		11	45.5% (5/11)	36.4% (4/11)	9.0% (1/11)
Time from the third vaccination to the analysis of serum (days)		222 (184–284)	236 (216–276)	219 (186–276)	208 (187–284)
COVID-19 (Breakthrough infection)		12.7% (8/63)	9.1% (1/11)	9.7% (3/31)	18.2% (4/21)
Duration until COVID-19 infection after the second vaccination (days)		146 (117–176)	176	135 (121–154)	135.5 (117–173)

TNF = tumor necrosis factor, 5-ASA = 5-aminosalicylic acid, CD = Crohn’s disease, UC = ulcerative colitis, UST = ustekinumab, IM = immunomodulator, AZA = azathioprine, 6-MP = mercaptopurine.

## Data Availability

The authors confirm that the data supporting the findings of this study are available within the article.

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
