# Peer review of "Impact of Anti-TNFα Treatment on the Humoral Response to the BNT162b2 mRNA COVID-19 Vaccine in Pediatric Inflammatory Bowel Disease Patients"

_vaccines, 2022, doi:10.3390/vaccines10101618_

Round 1
Reviewer 1 Report
This is a well-designed clinical study of COVID vaccination results with young inflammatory bowel diseases (IBD) patients on various immunosuppressive drugs. Humoral antibody titers were measured at different timepoints for the three groups of patients treated with anti-TNFα, non-anti-TNFα, and 5-ASA. The short-term adverse events were also documented. This is a comprehensive study and clearly crafted. A few minor comments are:
1. I would like to ask authors to point out in discussion that specific antibody titer in serum against SARS-2 spike protein dose not always directly translate to effective protection from virial infection, especially considering the ADE.
https://doi.org/10.1016/j.cell.2021.05.032
2. As mRNA immunization can stimulate immune responses for a long period of time. The potentially induced inflammation is worth for discussion in this study, especially for IBD patients.
3. As COVID vaccine associated AEs are merging, we must be careful when wording like “a third vaccination should be considered”.
https://www.cell.com/trends/molecular-medicine/fulltext/S1471-4914(22)00103-4
4. The second last row in Page 3: the sample size was 10 or 11?
Author Response
Dear Reviewer 1
Thank you for review in our paper entitled "Impact of anti-TNFα treatment on the humoral response to the BNT162b2 mRNA COVID-19 vaccine in pediatric inflammatory bowel disease patients". We have revised the text in accordance with the reviewer’s suggestions. All alterations made to our manuscript are shown in red in the revised version of the manuscript. I would like to thank the reviewers who had improved our paper to high quality matched Vaccines.
Reviewer #1:
- I would like to ask authors to point out in discussion that specific antibody titer in serum against SARS-2 spike protein dose not always directly translate to effective protection from virial infection, especially considering the ADE. https://doi.org/10.1016/j.cell.2021.05.032
Response: In considering SARS-CoV-2, it was necessary to mention ADE, so we added the sentences in lines 313-321 and 336-337 of Discussion, and references of [38] and [39] to discuss ADE in SARS-CoV-2.
- As mRNA immunization can stimulate immune responses for a long period of time. The potentially induced inflammation is worth for discussion in this study, especially for IBD patients.
Response: We added the sentence in lines 303-312 of Discussion, and a reference of [37] to mention that in a previous study of adult IBD patients, no exacerbation of IBD activity occurred after 6 months of the initial vaccination, same as our findings. However, we agree with the Reviewer that more long-term observational studies are needed regarding the exacerbation of IBD activity after vaccination.
- As COVID vaccine associated AEs are merging, we must be careful when wording like “a third vaccination should be considered”. https://www.cell.com/trends/molecular-medicine/fulltext/S1471-4914(22)00103-4
Response: We have changed the expression to "may" instead of "should" when we recommended vaccination.
- The second last row in Page 3: the sample size was 10 or 11?
We have changed it as your suggestion.
Response: We have changed 10 to 11.
Reviewer 2 Report
Many thanks for asking me to review this letter entitled "Impact of anti-TNFα treatment on the humoral response to the 2 BNT162b2 mRNA COVID-19 vaccine in pediatric inflammatory 3 bowel disease patients." dealing with the humoral response to the BNT162b2 vaccine and its efficacy in PIBD patients on immunosuppressive therapy, which is why I enjoyed reading this manuscript.
Authors presented a single-center, prospective, study just suffering the small sample size and the generalizability, but the authors largely and honestly discussed all these limitations.
The manuscript is well written and very pleasant to read.
I also liked the conclusion just limited to, and supported by the “balanced” results of the present study.
Just on Discussion, P7, Li245. “… the BNT162b2 vaccine.21” to change to “the BNT162b2 vaccine [21].”, if this is adapted.
Author Response
Dear Reviewer 2
Thank you for review in our paper entitled "Impact of anti-TNFα treatment on the humoral response to the BNT162b2 mRNA COVID-19 vaccine in pediatric inflammatory bowel disease patients". We have revised the text in accordance with the reviewer’s suggestions. All alterations made to our manuscript are shown in red in the revised version of the manuscript. I would like to thank the reviewers who had improved our paper to high quality matched Vaccines.
Reviewer #2:
Just on Discussion, P7, Li245. “… the BNT162b2 vaccine.21” to change to “the BNT162b2 vaccine [21].”, if this is adapted.
Response: We have changed it as your suggestion.
Reviewer 3 Report
In this manuscript the authors investigate the antibody response of PIBD patients to COVID19 vaccination as a function of anti-TNF antibody treatment. The results are consistent with some published studies in adults. The samples are analyzed at appropriate time points to demonstrate kinetics.
For the patient characteristics, there should be discussion of how long the patients were on treatment and whether this treatment was stable. Does this factor into any observed results?
Author Response
Dear Reviewer 3
Thank you for review in our paper entitled "Impact of anti-TNFα treatment on the humoral response to the BNT162b2 mRNA COVID-19 vaccine in pediatric inflammatory bowel disease patients". We have revised the text in accordance with the reviewer’s suggestions. All alterations made to our manuscript are shown in red in the revised version of the manuscript. I would like to thank the reviewers who had improved our paper to high quality matched Vaccines.
Reviewer #3:
For the patient characteristics, there should be discussion of how long the patients were on treatment and whether this treatment was stable. Does this factor into any observed results?
Response: We added the sentences in lines 119-123 of Result for the patient characteristics. In Table 1, we added a new column, "Time from the beginning of the IBD treatment to the second vaccination." Patients treated with 5-ASA had a significantly shorter time from the beginning of the IBD treatment to the second vaccination than those treated with non-anti-TNFα. However, since all patients in this study were in clinical remission at the time of analysis, the impact on the results was expected to be limited.